# Tracking down carbon inputs underground from an arid zone Australian calcrete

**Mattia Saccò**[1]*, **Alison J. Blyth**[1], **William F. Humphreys**[2,3], **Jen A. Middleton**[2], **Nicole E. White**[4], **Matthew Campbell**[1], **Masha Mousavi-Derazmahalleh**[4], **Alex Laini**[5], **Quan Hua**[6], **Karina Meredith**[6], **Steven J. B. Cooper**[7,8], **Christian Griebler**[9], **Sebastien Allard**[10], **Pauline Grierson**[2], **Kliti Grice**[1]

**1** WA-Organic Isotope Geochemistry Centre, The Institute for Geoscience Research, School of Earth and Planetary Sciences, Curtin University, Perth, WA, Australia, **2** School of Biological Sciences, University of Western Australia, Crawley, Western Australia, Australia, **3** Collections and Research Centre, Western Australian Museum, Welshpool, WA, Australia, **4** Trace and Environmental DNA Lab, School of Molecular and Life Sciences, Curtin University, Perth, WA, Australia, **5** Department of Chemistry, Life Sciences and Environmental Sustainability, University of Parma, Parco Area delle Scienze, Parma, Italy, **6** Australian Nuclear Science and Technology Organisation (ANSTO), Locked Bag Kirrawee DC, NSW, Australia, **7** Australian Centre for Evolutionary Biology and Biodiversity, School of Biological Sciences, University of Adelaide, South Australia, Australia, **8** Evolutionary Biology Unit, South Australian Museum, North Terrace, Adelaide, South Australia, Australia, **9** Department of Functional and Evolutionary Ecology, University of Vienna, Vienna, Austria, **10** Curtin Water Quality Research Centre, Curtin University, Perth, WA, Australia

* mattia.sacco@postgrad.curtin.edu.au

**Data Availability Statement:** All relevant data are within the manuscript and its Supporting Information files.

**Funding:** This research was funded by an Australian Research Council (ARC) linkage grant

## Abstract

Freshwater ecosystems play a key role in shaping the global carbon cycle and maintaining the ecological balance that sustains biodiversity worldwide. Surficial water bodies are often interconnected with groundwater, forming a physical continuum, and their interaction has been reported as a crucial driver for organic matter (OM) inputs in groundwater systems. However, despite the growing concerns related to increasing anthropogenic pressure and effects of global change to groundwater environments, our understanding of the dynamics regulating subterranean carbon flows is still sparse. We traced carbon composition and transformations in an arid zone calcrete aquifer using a novel multidisciplinary approach that combined isotopic analyses of dissolved organic carbon (DOC) and inorganic carbon (DIC) ($\delta^{13}C_{DOC}$, $\delta^{13}C_{DIC}$, $^{14}C_{DOC}$ and $^{14}C_{DIC}$) with fluorescence spectroscopy (Chromophoric Dissolved OM (CDOM) characterisation) and metabarcoding analyses (taxonomic and functional genomics on bacterial 16S rRNA). To compare dynamics linked to potential aquifer recharge processes, water samples were collected from two boreholes under contrasting rainfall: low rainfall ((LR), dry season) and high rainfall ((HR), wet season). Our isotopic results indicate limited changes and dominance of modern terrestrial carbon in the upper part (northeast) of the bore field, but correlation between HR and increased old and $^{13}C$-enriched DOC in the lower area (southwest). CDOM results show a shift from terrestrially to microbially derived compounds after rainfall in the same lower field bore, which was also sampled for microbial genetics. Functional genomic results showed increased genes coding for degradative pathways—dominated by those related to aromatic compound metabolisms—during HR. Our results indicate that rainfall leads to different responses in different parts of the bore field, with an increase in old carbon sources and microbial

(LP140100555) to the University of Adelaide, Curtin University, and Flinders University, with industry partners, the Western Australian Museum, the South Australian Museum, Rio Tinto, Biota Environmental Sciences, Bennelongia Environmental Consultants and the Department of Parks and Wildlife (WA). The Environment Institute and School of Biological Sciences, University of Adelaide, funded a lease of the Sturt Meadows calcrete bore field. MS is supported by a Curtin International Postgraduate Research Scholarship (CIPRS) and an AINSE postgraduate scholarship (PGRA). AB acknowledges an AINSE Research Fellowship (2012-2018). We acknowledge financial support from the Australian Government's National Collaborative Research Infrastructure Strategy (NCRIS) for the Centre for Accelerator Science at the Australian Nuclear Science and Technology Organisation. The authors declare that they have no conflicts of interest.

**Competing interests:** The authors have declared that no competing interests exist.

processing in the lower part of the field. We hypothesise that this may be due to increasing salinity, either due to mobilisation of $Cl^-$ from the soil, or infiltration from the downstream salt lake during HR. This study is the first to use a multi-technique assessment using stable and radioactive isotopes together with functional genomics to probe the principal organic biogeochemical pathways regulating an arid zone calcrete system. Further investigations involving extensive sampling from diverse groundwater ecosystems will allow better understanding of the microbiological pathways sustaining the ecological functioning of subterranean biota.

## Introduction

The global carbon cycle fuels the processes that are responsible for maintaining the ecological functioning of ecosystems [1,2]. Terrestrial environments, together with oceans, play a key role in sequestering atmospheric carbon pools and allow fundamental recycling of biomass [3]. However, on-going global warming, mainly caused by increased greenhouse gases linked with anthropogenic activities, is putting at risk the maintenance of this ecological balance [4,5].

During the last decade, carbon storage and fluxes in freshwater environments have gained prominence as key factors in the global cycling of organic matter [6,7,8]. Drake et al. [9] estimated up to 5.1 Pg y$^{-1}$ of carbon delivered from land to surficial inland aquatic systems (lakes, rivers, reservoirs). Kayranli et al. [10] reported that soil and sediment from wetlands are amongst the world's most extensive carbon sinks, with peatlands accounting for a third of the organic soil worldwide [11]. These observations are in concordance with Keiluweit, et al. [12], who indicated that surficial soil and unsaturated zones provide the biggest carbon source within the terrestrial framework. However, while widely investigated in surficial ecosystems, carbon flows are understudied in groundwater environments [13,14].

Groundwater systems are often hydrologically interconnected with each other and/or to surface terrestrial environments and water bodies [15]. Especially in arid environments, near-surface groundwaters (e.g. groundwater dependent ecosystems (GDEs)), provide a vital conceptual and physical continuum [16]. Surface water-groundwater exchanges (SW/GW) shape biogeochemical dynamics, including carbon cycling and nutrient circulation, which regulate the functioning of both surface and subterranean ecosystems [17]. However, dissolved carbon concentrations within aquatic subterranean environments are typically orders of magnitude lower than lakes and rivers [18,19,20]. McDonough et al. [21] reported average subterranean global dissolved organic concentrations (DOC) of ~ 1 mg L$^{-1}$ f, while the global flux of inorganic carbon (DIC) into groundwater is estimated to be 0.2 GtC y$^{-1}$ [22].

Subterranean DOC replenishment can occur either *via* SW/GW and/or *via* rainfall recharge through soils containing high organic matter (OM) content [23,24]. Baker et al. [25] suggested that seasonal saturation of sediments overlying unconfined groundwater plays a key role in regulating organic carbon dynamics underground. Similar site-specific models have been suggested over the last two decades [26,27], emphasising the importance of the vadose zone as a source of carbon for groundwater biological communities and biogeochemical cycling [28].

Groundwater communities are thought to be bottom-up regulated by the availability of OM which drives ecological functioning (i.e. energy flows, trophic cascade effects) in groundwater ecosystems [29]. Microbial diversity and productivity are considered to be limited by the concentration and bioavailability of OM in groundwater [30]. While heterotrophic

metabolism is commonly considered a major process for sustaining food web interactions in a typically low-energy system [31], chemolithoautotrophic strategies have also been extensively reported [32,33]. Microbially-processed OM, together with detrital fractions [34], are transferred to higher trophic levels of subterranean biota [35] by higher primary consumers (i.e. terrestrial (troglofauna) and obligate aquatic (stygofauna)). As a result, subterranean carbon turnovers, often linked with recharge regimes [36], are ultimately responsible for cascading effects on energy flows and food web interactions [37].

The Sturt Meadows (SM) calcrete in Western Australia hosts a stygofaunal community composed of 18 macroinvertebrate species including blind dytiscid beetles and chiltoniid amphipods, and is a hotspot for subterranean aquatic invertebrate diversity [38,39]. Recent investigations into the ecological functioning of the calcrete stygofaunal assemblages [40,41] have indicated that rainfall input dynamics play a vital role in shaping cascade effects. Here, we extend this research by investigating carbon input dynamics and microbial processing under contrasting rainfall periods *via* hydrochemistry, stable and radiocarbon isotope ecology and DNA metabarcoding analyses. Through this multidisciplinary approach, we aim to 1) elucidate the nature of the carbon inputs under differential rainfall regimes, 2) provide isotope-based tracking of the organic and inorganic carbon sources in the groundwater, and 3) identify metabolic and functional microbial patterns coupled with organic inflows linked to rainfall percolation. The study of carbon inputs and their microbial incorporation has the potential to expand our understanding of the ecological dynamics sustaining biodiversity in this taxonomic hotspot.

## Methodology

### Study area

Field work was carried out at the Sturt Meadows (SM) calcrete, located within Sturt Meadows pastoral station in the northeast of the Yilgarn region (28˚41′S 120˚58′E), Western Australia (Fig 1A). The Yilgarn craton is one of the most important Late Archaean metallogenic provinces in the world [42], and constitutes the bulk of this Western Australian region. The area hosts calcretes formed by the precipitation of calcium carbonate along palaeodrainage channels [43], which have been the focus of research for more than a century [44,45,46]. The SM aquifer is located upstream of Lake Raeside, covering an area of ~43 km$^2$, and has a strong biogeochemical gradient comparable to estuarine systems [39]. Previous studies of the depth and lithography of the calcrete [47,48] identified two geological sectors: calcretes and clayey formations (Fig 1B). The mean permeability of the SM calcrete is similar to that of sand (1.9–4.6 × 10$^{-4}$ m s–1 [49]), suggesting an average porosity of ~25% [50]. The average yearly rainfall of the area is low, at around 200 mm, and pan evaporation is 2400 mm year$^{-1}$ (BoM). The aquifer is very shallow, located two to four metres below the surface, and is accessible through boreholes, initially drilled for mineral exploration, along a grid that can be divided into two sections. The northern grid is 0.9 × 1.4 km with bores spacing at 100 m in north-south and east-west directions, while the southern section is 1.2 × 0.9 km, containing bores separated from each other 100 m east–west and 200 m north–south (Fig 1B). These bores are unlined, except for about the upper 0.5 m which are lined with 10 cm diameter PVC pipe for stability, and capped with PVC lids to avoid rainfall falling directly into the aquifer [50].

### Field work procedures and sample preparation

Groundwater samples were collected from the unlined bores at D13 (zone CD) and W4 (zone A2) using a submersible centrifugal pump (GEOSub 12V Purging Pump) after three well-volumes were purged and stabilisation of in-field parameters (temperature, pH, salinity, dissolved

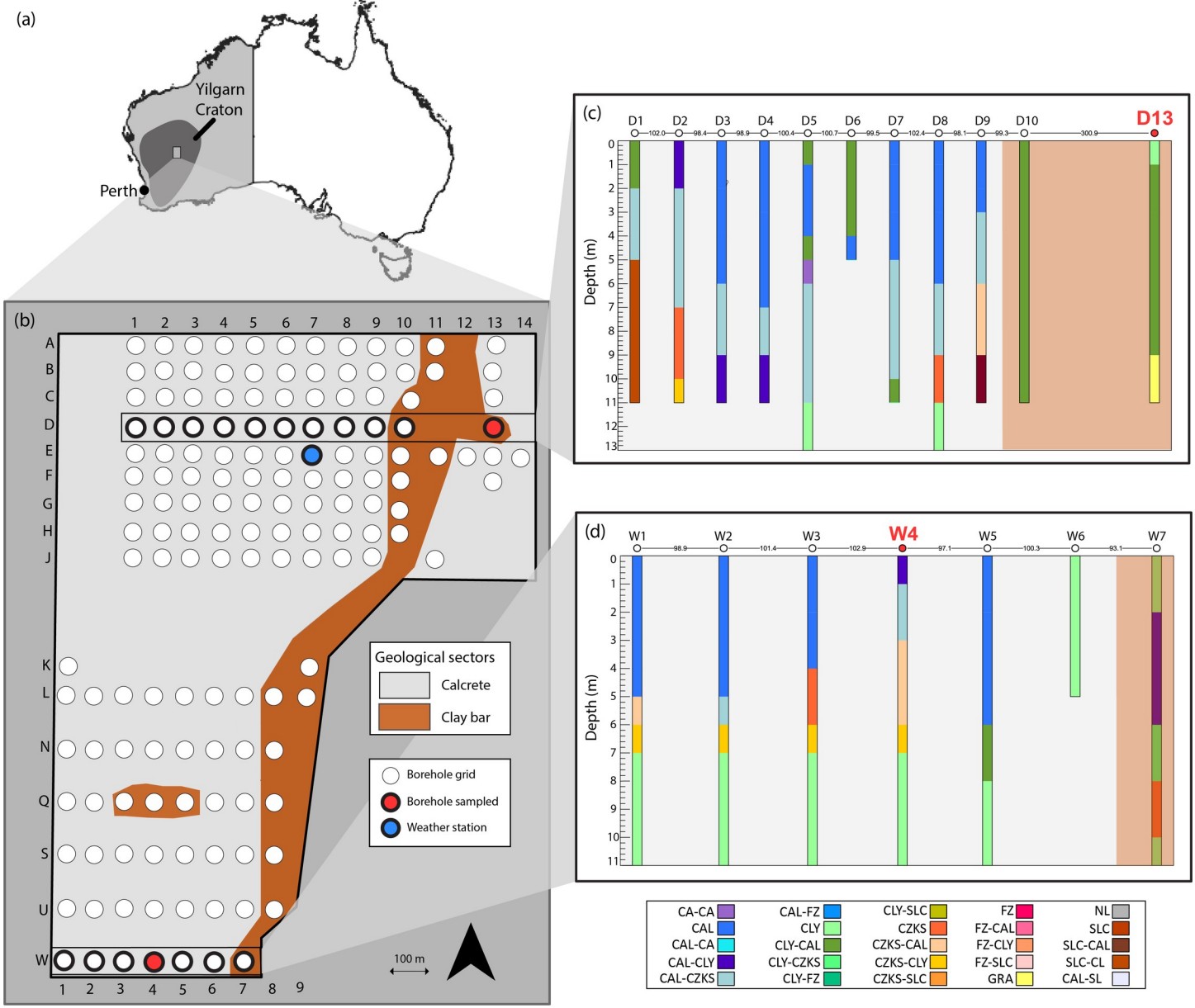

**Fig 1.** Borehole grid showing its location within the Yilgarn region (a), the geological sectors (b) and the bores sampled (D13 and W4, in red), together with the lithological profiles ((c) and (d)). CAL: soft calcrete, CZKS: siliceous calcrete, FZ: ferruginous zone, CLY: clay, GRA: granite, NL: no geology log, SLC: silcrete, GDR: granodiorite, ASB: asbestos, CA: cavity.

oxygen (DO) and oxidation-reduction potential (ORP)) was observed. The selected bores are representative of the two main geological units of the area, W4 being in calcrete and D13 in clayey formations (Fig 1B, 1C and 1D). Preliminary investigations on the hydrology of the SM aquifer indicated that these two bores are the most reliable (i.e. lowest risk of drying) to test biogeochemical and ecological patterns across time [41].

Rainfall and groundwater level fluctuations were monitored for one year (from 18/06/2017 to 17/06/2018) through a weather station installed near bore E7 (Fig 1B), and indicated very patchy rainfall events and periodic recharge dynamics typical of Western Australian calcrete systems [39]. However, monitoring of groundwater chemistry at the SM calcrete revealed

infiltration of rainfall from the surface together with increased inputs of ammonia after rain precipitation [41]. For this study, two sampling campaigns, corresponding to contrasting rainfall periods as categorised by Hyde et al. [51], were carried out on the 7/11/2017 (low rainfall, LR; <10 mm of rain during the 30 days prior to sampling) and on the 17/03/2018 (high rainfall, HR; >30 mm of rain during the 30 days prior to sampling).

Changes in carbon content in water after different levels of rainfall were investigated using dissolved organic (DOC) and inorganic (DIC) carbon and their isotopes ($\delta^{13}C_{DOC}$ and $\delta^{13}C_{DIC}$) coupled with radiocarbon analysis ($^{14}C_{DOC}$ and $^{14}C_{DIC}$). These techniques were complemented by measuring the DOC fluorescence. Samples for $\delta^{13}C_{DOC}$ were filtered through 0.2 µm glass fiber filters, collected in 60 mL HDPE bottles and frozen after sampling. The $^{14}C_{DOC}$ samples were filtered through 0.2 µm filters, collected in 1 L HDPE bottles and frozen after sampling. The $\delta^{13}C_{DIC}$ samples were filtered through 0.2 µm filters, collected in 12 mL glass vials (Exetainers) and refrigerated after sampling. Samples for $^{14}C_{DIC}$ analysis were filtered through 0.45 µm filters and collected in 1 L HDPE, with no further treatment. The DOC fluorescence samples were collected in 1 L HDPE bottles and kept refrigerated in darkness until further tests. Other hydrochemistry parameters such as water isotopes ($^{3}H$, $\delta^{18}O$ and $\delta^{2}H$) and chloride concentration (Cl⁻) were measured in water samples collected in 1 L HDPE bottles that were immediately frozen until further analyses. All samples were sealed with sealing tape after collection to limit atmospheric exchange, and kept in the dark.

Water samples for functional genomic investigations on the microbial community were collected from the bore W4 and stored in 1 L HDPE bottles and frozen immediately after collection. Samples were then filtered through 0.4 µm nitrocellulose membrane filters (Millipore, Sigma, Burlington, MA, USA) using a vacuum system, and the filtered content was kept frozen (-20˚C) until further analyses. Temperature, pH, ORP, salinity, DO and depth were measured *in situ* (bores D13 and W4) using portable field measurement equipment (Hydrolab Quanta Multi-Probe Meter®). The field site was accessed and samples were collected with permit approval (permit number 08-003150-1) from the Department of Parks and Wildlife of Western Australia.

## Instrument methods and data analysis

**Biogeochemical measurements.** DOC was determined by the non-purgeable organic carbon (NPOC) method using a Shimadzu high temperature combustion TOC-L/TNM-L analyser. DIC was obtained through a total organic carbon (TOC) configuration which measured the total carbon, followed by inorganic carbon. The TOC analysis was based on a standard method 5310-B [52] with detection by NDIR detector. Both DOC and DIC analyses were run in duplicates and the combustion temperature was 720˚C.

$\delta^{13}C_{DOC}$ isotopic ratios of waters were calculated using a Liquid Chromatography Isotope Ratio Mass Spectrometer (LC-IRMS) at the La Trobe Institute for Molecular Sciences (LIMS, La Trobe University, Melbourne, Australia) composed by a Accela 600 pump connected to a Delta V Plus Isotope Ratio Mass Spectrometer via a Thermo Scientific LC Isolink (Thermo Scientific). $\delta^{13}C_{DIC}$ isotopic ratios in water were analysed by Isotope Ratio Mass Spectrometer–Western Australia Biogeochemistry Centre at The University of Western Australia using a GasBench II coupled with a Delta XL Mass Spectrometer (Thermo-Fisher Scientific)—and results, with a precision of ± 0.10 per mil (‰), were reported as ‰ deviation from the NBS19 and NSB18 international carbonate standard [53]. $\delta^{13}C_{DOC}$ and $\delta^{13}C_{DIC}$ values were reported ‰ relative to the Vienna Peedee Belemnite (VPDB).

For radiocarbon analyses of both the DOC and DIC ($^{14}C_{DOC}$ and $^{14}C_{DIC}$), pre-treated samples were subjected to $CO_2$ extraction and graphitization following the methodology of Hua

et al. [54] and Bryan et al. [55]. $^{14}$C content of samples was determined by means of the Accelerator Mass Spectrometry (AMS) at ANSTO (Australian Nuclear Science and Technology Organisation) in Sydney, Australia [56]. Radiocarbon results were reported in conventional age before present (BP, with BP being 1950), percent of modern carbon (pMC) and $\Delta^{14}$C value in per mil (‰) relative to the absolute radiocarbon standard activity in 1950 [57].

Absorbance scans and excitation emission matrices (EEMs) were recorded simultaneously using an Aqualog® (Horiba Scientific). Fluorescence intensities were measured at excitation wavelengths 250–500 nm (1 nm increments) and emission wavelengths 250–575 nm (3 nm increments). The composition of DOM was characterised by a range of indices (HIX$_{EM}$, BIX, FI, SUVA254; S2 Table) and by identifying individual fluorescent components using parallel factor analysis (PARAFAC) [58].

The $\delta^{18}$O and $\delta^2$H were analysed by IRMS at ANSTO, and their values are reported as per mil (‰) deviations from the international standard V-SMOW and were reproducible to ±0.1‰ and ±1.0‰. The $^3$H activities were expressed in tritium units (TU, uncertainty of $\pm \leq$ 0.1 TU and quantification limit of $\leq$ 1 TU) and were analysed by liquid scintillation counting [59].

**Genetic analyses.** Three 1 litre water sample replicates collected from bore W4 (zone A2) during both rainfall periods (LR and HR) were used for bacterial 16S metabarcoding and microbial functional analysis. Water samples were filtered using two Sentino peristaltic microbiology pumps (Pall Life 126 Sciences, New York, USA), through 0.45 μm sterile membrane filters (Pall Life Sciences, New York, USA). All water filtering equipment was soaked for a minimum of 10 minutes in 10% sodium hypochlorite solution and treated with UV light prior to use and between samples. Immediately post-filtering, half of the filter membrane was used for DNA extraction, while the remaining half was frozen at -20˚C.

Water membranes, inclusive of laboratory controls, were extracted using DNeasy Blood and Tissue Kit (Qiagen; Venlo, Netherlands), with the following modifications to the manufacturer's protocol. For the DNA digest, both the ATL buffer (360 μL) and Proteinase K (40 μL) solutions were doubled to ensure that the membranes were adequately exposed to the lysis solution to optimise DNA yield. The DNA digests were incubated (56˚C) overnight in a rotating hybridisation oven. The digest was transferred into a clean tube and loaded into a QIAcube (Qiagen; Venlo, Netherlands) automated DNA extraction system for the remainder of the extraction process. The DNA was eluted off the silica column in 100 μL AE buffer.

The quality and quantity of DNA extracted from each water membrane was measured using quantitative PCR (qPCR), targeting the bacterial 16S gene. PCR amplifications to assess the quality and quantity of the DNA target of interest via qPCR (Applied Biosystems [ABI], USA) were carried out in 25 μL reaction volumes consisting of 2 mM MgCl2 (Fisher Biotec, Australia), 1 x PCR Gold Buffer (Fisher Biotec, Australia), 0.4 μM dNTPs (Astral Scientific, Australia), 0.1 mg bovine serum albumin (Fisher Biotec, Australia), 0.4 μM of each primer (Bact16S_515F and Bact16S_806R; [60,61]), 0.2 μL of AmpliTaq Gold (AmpliTaq Gold, ABI, USA) and 2 μL of template DNA (Neat, 1/10, 1/100 dilutions). The cycling conditions were: initial denaturation at 95˚C for 5 minutes, followed by 40 cycles of 95˚C for 30 seconds, 52˚C for 30 seconds, 72˚C for 30 seconds, and a final extension at 72˚C for 10 minutes.

Extracts that successfully yielded DNA of sufficient quality, free of inhibition, as determined by the initial qPCR screen (detailed above), were assigned a unique 6-8bp multiplex identifier tag (MID-tag) with the bacterial 16S primer set. Independent MID-tag qPCR for each water membrane were carried out in 25μL reactions containing 1X PCR Gold Buffer, 2.5mM MgCl2, 0.4mg/mL BSA, 0.25mM of each dNTP, 0.4μM of each primer, 0.2μL AmpliTaq Gold and 2–4μL of DNA as determined by the initial qPCR screen. The cycling conditions for qPCR using the MID-tag primer sets were as described above. MID-tag PCR amplicons were

generated in duplicate and the library was pooled in equimolar ratio post-PCR for DNA sequencing. The final library was size selected (160-600bp) using Pippin Prep (Sage Sciences, USA) to remove any MID-tag primer-dimer products that may have formed during amplification. The final library concentration was determined using a QuBitTM 4 Fluorometer (Thermofischer, Australia) and sequenced using a 300 cycle V2 kit on an Illumina MiSeq platform (Illumina, USA).

MID-tag bacterial 16S sequence reads obtained from the MiSeq were sorted (filtered) back to the water sample based on the MID-tags assigned to each DNA extract using Geneious v10.2.5 [62]. MID-tag and primer sequences were trimmed from the sequence reads allowing for no mismatch in length or base composition.

Filtered reads were input into a containerised workflow comprising USEARCH [63] and BLASTN [64]. The fastx-uniques, unoise3 (with minimum abundance of 8) and otutab commands of USEARCH were applied to generate unique sequences, ZOTUs (zero-radius OTUs) and abundance table, respectively. The ZOTUs were compared against the nucleotide database using the following parameters in BLASTN: perc_identity $> = 94$, evalue $< = 1e-3$, best_hit_score_edge 0.05, best_hit_overhang 0.25, qcov_hsp_perc 100, max_target_seqs = 5. An in-house Python script was used to assign the ZOTUs to their lowest common ancestor (LCA). The threshold for dropping a taxonomic assignment to LCA was set to perc_identity $> = 96$ and the difference between %identity of the two hits when their query coverage is equal was set to 1.

To investigate functional activity involved in carbon cycling, the 16S metabarcoding data were fed to the Phylogenetic Investigation of Communities by Reconstruction of Unobserved States 2 (PICRUSt2) software package to generate predicted metagenome profiles [65]. These profiles were clustered into Kyoto Encyclopedia of Genes and Genomes (KEGG) Orthologs (KOs) [66] and MetaCyc pathway abundances [67] focusing on carbon metabolism and degradation pathways, respectively.

## Statistical analysis

The statistical analyses on isotope, fluorescence and absorbance data were performed in R software version 3.6.0 (Development-Core-Team, 2016). DOC, DIC, $\delta^{13}C_{DOC}$ and $\delta^{13}C_{DIC}$ values (obtained from two independent replicates per parameter) per bore (W4 and D13) were compared across the two rainfall events using ANOVAs (R-package 'stats'). Radiocarbon results were unique per bore and sampling campaign, therefore data were not analysed statistically.

The R package staRdom (version 1.1.1) [68] was used to correct EEMs, calculate all fluorescence/absorbance indices and for conducting PARAFAC modelling. EEMs were corrected for blanks (Milli-Q water), inner filter effects, Raman normalised [69], and scatter (Raman and Rayleigh) were removed and interpolated prior to PARAFAC. Our PARAFAC model was split-half validated [70] and recognized five fluorescent components (S1 Fig). These components are reported as maximum fluorescence intensity of each component (Fmax) in each sample. Principal Components Analysis (PCA) was conducted on fluorescence/absorbance indices to assess differences between sites and rainfall period. The R studio (version 3.6.1) 'prcomp' function was used to carry out the PCA and results are presented in two dimensions (PC1 and PC2) along with eigenvectors. Differences in $HIX_{EM}$, BIX, FI and $A_{254}$ between sites and rainfall periods were tested using 2-way ANOVA, where site and rainfall period (and their possible interaction) were treated as fixed factors. Tukey's HSD tests were performed to determine which of the means were significantly different when significant main effects were found. Data were log transformed to achieve normality when required.

Beta diversity patterns—the variations in species composition among rainfall periods—were analysed through the calculation of the Ochiai index [71] (R-package 'adespatial') and the quotient of the temporal turnover (Simpson pairwise dissimilarity) and total dissimilarity (measures as Sorensen pair-wise dissimilarity) (R-package 'betapart', function *beta.pair*). Pielou's evenness index (J) was calculated to infer the degree of dominant species in abundance, with values ranging from 0 (no evenness at all) to 1 (complete evenness). The Phyloseq package in R [72] was used to plot the ZOTU abundance at the family and genus level for low rainfall (LR) and high rainfall (HR) periods from the bore W4. The Statistical Analysis of Metagenomic Profiles (STAMP) bioinformatics software package was used to visualize and determine statistically significant results from the PICRUSt2 output [73]. For comparison of potential microbial metabolic shifts across rainfall periods, the White's non parametric t-test was used for both carbon metabolism and degradation pathways with confidence intervals of 95%, and visualized with extended error bar plots.

## Results

### Carbon inputs across rainfall periods

The DOC concentrations ranged from 0.39 ± 0.21 mg/L (W4 under LR) to 1.94 ± 0.75 mg/L (D13 during HR), while concentrations of dissolved inorganic carbon (DIC) ranged from 63.5 ± 0.14 mg/L (W4 under LR) to 87.44 ± 0.66 mg/L (D13 during HR). D13 had consistently higher DOC than W4.

DOC concentrations for bore W4 significantly increased under HR compared to LR (ANOVA, $P < 0.05$). Concurrently, W4 had significantly more positive $\delta^{13}C_{DOC}$ values under HR conditions than under LR conditions (ANOVA, $P < 0.001$) (Fig 2A). Groundwater from bore D13 also had higher DOC levels under HR (Fig 2A) than those during LR, but this was not statistically significant. $\delta^{13}C_{DOC}$ values in D13 did not change after rainfall (S1 Table), while its $^{14}C_{DOC}$ ages also remained similar between LR and HR, being younger than those in W4 (Fig 2C). Compared to W4, DIC concentrations were higher in D13, and $\delta^{13}C_{DIC}$ values were less enriched, but these differences were not statistically significant (Fig 2B). Similar increasing trends were found for DIC concentration and $\Delta^{14}C_{DIC}$ values for the two bores when LR was compared to HR (Fig 2D).

Increasing trends for water temperature, DO (bore D13) and chloride concentrations where coupled with decreasing patterns for pH, DO (bore W4), ORP and depth (S4 Table) when LR is compared to HR. $\delta^{18}O$ and $\delta^{2}H$ values did not vary across rainfall periods within the two bores analysed, while Tritium values from bore D13 were slightly lower during HR (0.53TU) when compared to LR (0.77 TU).

### Fluorescence and absorbance characterisation

Parallel factor analysis (PARAFAC) identified five unique humic-like fluorescent components (S1 Fig). Component 1 (C1) had a primary excitation peak at <250 nm and secondary peak at 330 nm with a broad emission peak from 370 to >575 nm (Em. max at 415 nm). Component 2 (C2) had an excitation peak at <250 nm and at 300 nm with a broad emission peak from 350 to >575 nm (Em. max at 395 nm). Component 3 (C3) had an excitation peak at 268 nm and at 386 nm with a broad emission peak from 400 to >575 nm (Em. max at 446 nm). Component 4 (C4) had an excitation peak at 260 nm and at 370 nm with a broad emission peak from 420 to >600 nm (Em. max at 493 nm). Component 5 (C5) had an excitation peak at 250 nm and at 318 nm with an emission peak from 310 to >410 nm (Em. max at 364 nm). The rainfall affected the fluorescence intensity of all PARAFAC components. For site D13, the fluorescence maximum (Fmax) of all components increased after HR, while site W4 displayed the opposite

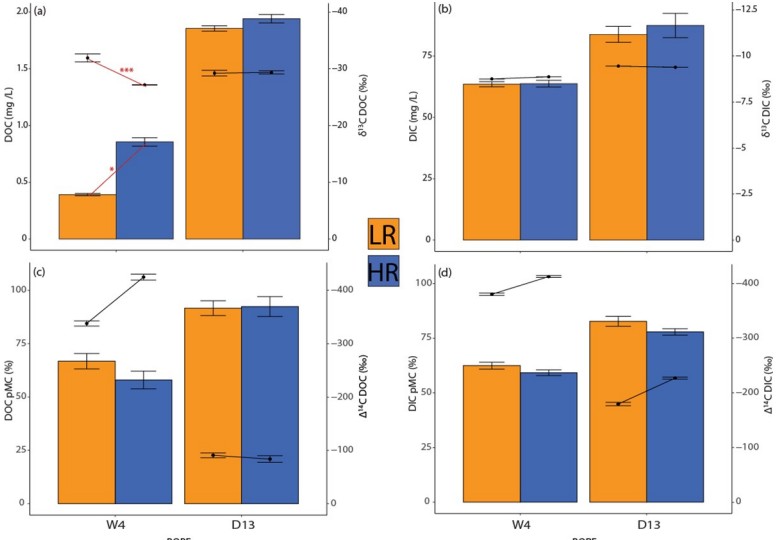

**Fig 2.** Bar charts illustrating DOC and DIC concentrations (a and b), and respective percent of modern carbon (pMC, with modern defined as 1950) (c and d) from the bores W4 and D13 across LR (dark yellow) and HR (blue). The whiskers of the bars refer to standard deviation values. Combined line graphs refer to $\delta^{13}C$ DOC (a), $\delta^{13}C$ DIC (b), $\Delta^{14}C$ DOC (c) and $\Delta^{14}C$ DIC (d). * significant trend with $P$ value < 0.05; ** significant trend with $P$ value < 0.005; *** significant trend with $P$ value < 0.0005. Refer to S1 Table for the specific values of the parameters.

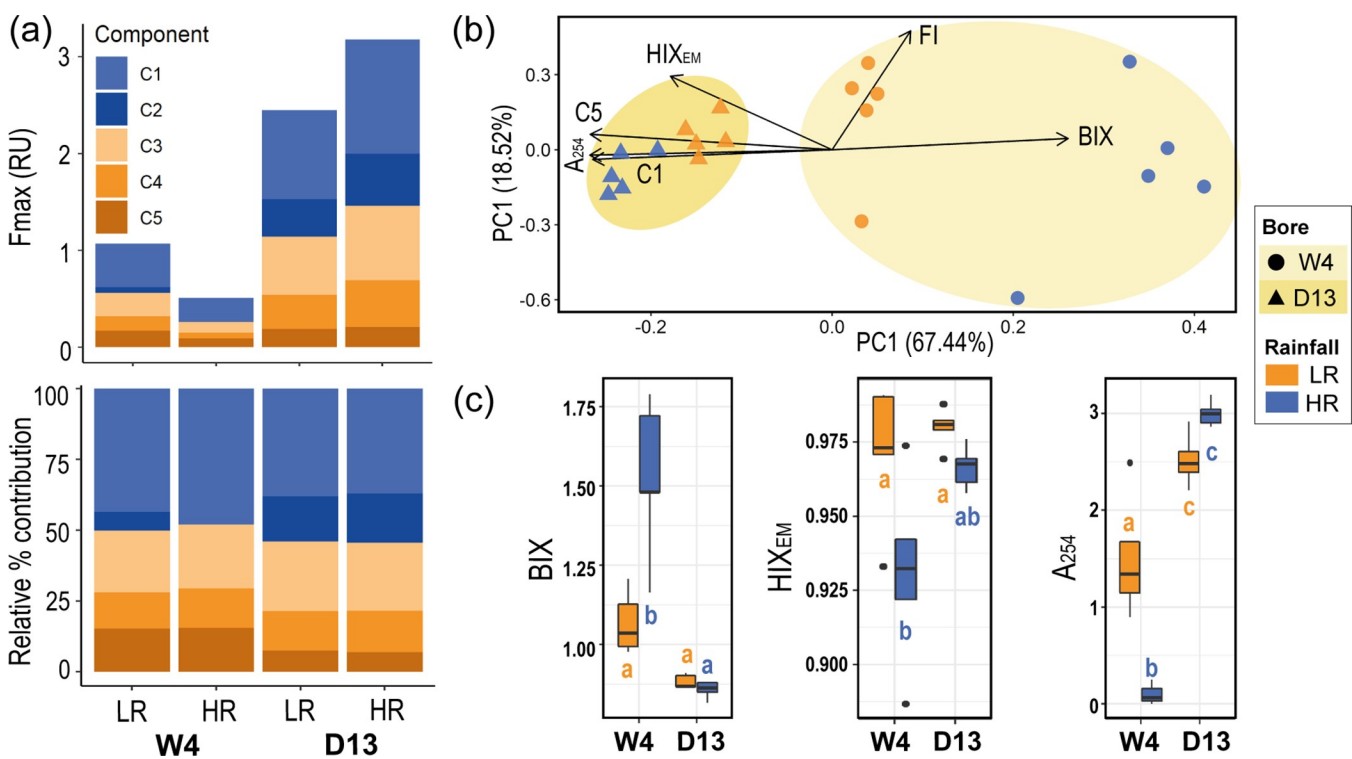

**Fig 3. CDOM characterisation between bores and recharge periods.** (a) Fluorescence maximum in Raman units (Fmax) and percent contribution of the five PARAFAC components (i.e. C1, C2, C3, C4 and C5), (b) PCoA ordination discriminating PARAFAC components, fluorescence indices ($HIX_{EM}$, BIX) and absorbance indices ($A_{254}$) where correlations of indices with axes are visualised as arrows, and, (c) BIX, $HIX_{EM}$, and $A_{254}$ values. Significant post hoc comparisons ($P < 0.05$) are indicated by lowercase letters.

trend, with elevated fluorescence after low rainfall (LR) (Fig 3A). During both rainfall periods the Fmax of all components at site D13 was greater compared to site W4 (Fig 3A). Further, the relative composition of components changed between bores. C1 was most predominant across both bores and recharge periods explaining 37–50% of the CDOM signal. The contribution of C3 and C4 was consistent across samples and rainfall regimes ranging from 20–25% and 12–15% respectively. C5 had the largest change in contribution between the bores; contributing 7–8% at bore D13 and 13–18% at bore W4. Finally, during HR there was no presence of C2 at bore W4 (Fig 3A).

Optical indices (HIX$_{EM}$, A$_{254}$, BIX) varied between sites and rainfall period (Fig 3B). Overall, PCA of optical indices revealed a marked shift in CDOM composition for site W4, from more terrestrially derived compounds during LR to compounds with a lesser degree of humification during HR (i.e., microbial derived, autochthonous) (Fig 3B). In contrast, site D13 displayed negligible changes in CDOM composition, displaying slightly greater intensity of humic-like/terrestrial compounds during HR compared to during LR (Fig 3B). The humification index (HIX$_{EM}$) ranged from 0.89 to 0.99, indicating that CDOM for both bores and rainfall periods was largely comprised of humic compounds, as HIX$_{EM}$ values above 0.9 indicate a greater degree of humification [75,76,77]. During HR, both sites showed a marginal decrease in their HIX$_{EM}$ values, especially for site W4, however both remained close to 0.9 (Fig 3C). Greater A$_{254}$ absorbance at bore D13 indicated more aromatic content than site W4 (Fig 3B and 3C). Interestingly, BIX was greater at site W4 (μ = 1.53 ± 0.11) during HR than during LR (μ = 1.07 ± 0.04) and compared to site D13 (μ = 0.87 ± 0.01) for either rainfall period ($P < 0.05$, Fig 3C). The fluorescence index (FI) indicated CDOM was of terrestrial origin (FI ~ 1.4; [78]) and did not change between bores or rainfall periods (μ = 1.46 ± 0.02; Fig 3C).

## Microbial patterns

The 16S rRNA sequencing yielded 7503 sequences clustered into 87 ZOTUs (37 ZOTUs either belonged to uncultured bacteria or no reference was available). After the removal of the ZOTUs associated with the lab controls (N = 16), 25 ZOTUs were unique to LR, 25 ZOTUs belonged to HR, and both rainfall periods shared the other 21 ZOTUs. During LR, the dominant ZOTUs belonged to the families Rhodobacteraceae (*Paracoccus* sp. and *Roseivarius* sp.), Pseudomonadaceae (*Pseudomonas* sp.), Planococcacea (*Planomicrobium* sp.) and Caulobacteraceae (*Brevundimonas* sp.). Under HR the dominant ZOTUs corresponded to the families Rhodobacteraceae (*Stappia* sp. and *Roseibacterium* sp.), Phyllobacteriaceae (*Nitratireductor* sp.) and Rhodospirillaceae (*Thalassobaculum* sp. and *Tageae* sp.) (S2 Fig). All the genera experienced turnovers between LR and HR (Ochiai index, $P < 0.05$), suggesting that a shift in community assemblages across the two rainfall events had occurred. Specifically, 81.5% of the dissimilarity is due to genus replacement between rainfall periods (turnover), with the rest (18.5%) explained by the nestedness (species loss from rainfall period to rainfall period). Values of the Pielou's evenness index (J) during LR and HR ranged from 0.71 to 0.74.

Predictions of the quantitative proportion of individual metabolic pathways related to carbon turnover revealed a dominance of carbon fixation (46%) and methane metabolism (40%), followed by carbohydrate (8%) and lipid metabolisms (6%) (Fig 4A). Despite being more abundant under LR for the former two, none of the four main metabolic categories changed significantly between LR and HR. Pairwise tests indicated that 4 out of the 10 carbon processing pathways (Fig 4B) and 10 out of the 76 degradative pathways examined (Fig 4C) were significantly ($P < 0.05$) overrepresented in one of the two rainfall periods (either LR or HR). For carbon metabolism, the dicarboxylate-hydroxybutyrate cycle was more abundant during LR, whereas the reductive pentose phosphate cycle, pentose phosphate pathway and reductive

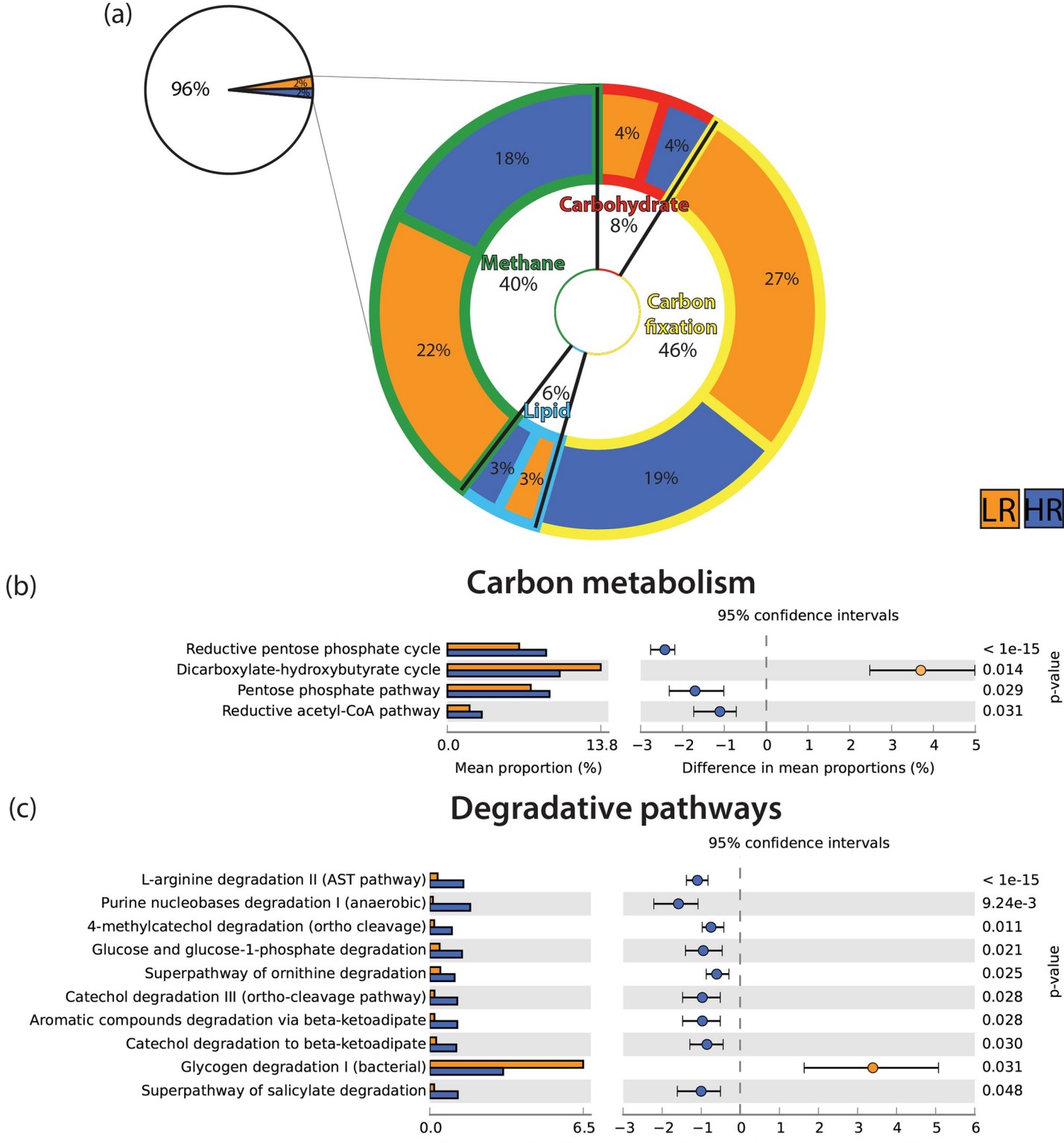

**Fig 4. Prediction of the microbial community metabolic status based on 16S rRNA amplicon sequencing and functional genomics analyses between LR and HR periods from the bore W4.** (a) doughnut chart showing the proportion of the metabolisms considered compared with the total pathways detected and the specific proportions of methane (green), carbohydrates (red), lipid (light blue and carbon fixation (yellow) metabolisms (derived from KOs). (b) and (c) extended error bar plots of predictive metagenome pathways differentially abundant between rainfall periods (*P* < 0.05, White's non parametric t-test).

acetyl-CoA pathway were more abundant during HR. With the exception of the glycogen degradation pathway, all the degradative pathways (arginine, purine, catechol, glucose, salicylate and aromatic compounds) were more abundant during HR. All pathways tested can be found in S3 Table.

## Discussion

### Carbon replenishment in groundwater systems

In groundwater systems, carbon is replenished by diffuse recharge through the unsaturated zone and/or *via* direct recharge from surface waters [79]. These processes are linked to rainfall conditions (i.e. wet/dry periods) and the hydrology of the system. SW/GW interactions drive OM incorporation into the ecosystem, which is typically characterised by low carbon content [21].

Aquifer recharge indicators such as tritium, oxygen-18 and deuterium did not vary much in the SM system between rainfall periods, suggesting limited recharge during our study. Conversely, chloride concentrations increased under HR (S4 Table), suggesting intrusion of hypersaline water from the surface during this period. These results indicate that carbon and nutrient inflows occur despite low recharge after rainfall, suggesting that soil zone processing plays a key role in regulating the biochemical flows at SM aquifer [25].

DOC concentrations show increasing trends after rainfall (HR), although only statistically significant for W4, indicating some carbon inputs to the system. At bore W4, older ($^{14}C_{DOC}$) and enriched DOC ($\delta^{13}C_{DOC}$) was found under HR, suggesting a sedimentary organic matter source, likely subject to microbial reprocessing causing stable isotope enrichment. In contrast, bore D13 showed stable trends characterised by modern DOC inputs. This difference in biochemical patterns suggests that *in situ* carbon sources play a central role at bore W4, possibly in tandem with changes in microbial activity occurring during HR. Meanwhile, bore D13 is receiving steady inflows of younger (and less microbially recycled) OM.

Patterns of DIC concentrations and $\delta^{13}C_{DIC}$ were steady across rainfall regimes. Inorganic dissolution was higher in D13, and input from younger carbonates (the inorganic fraction of carbon in calcretes) was detected. Overall, our isotopic data from organic and inorganic carbon indicated different responses in the upper (northeast, D13) and lower (southwest, W4) catchments, with D13 showing more modern carbon but less response to the rainfall event.

Groundwater CDOM quality depended on the bore and rainfall period. Humification ($HIX_{EM}$) and fluorescence index (FI) values indicated that CDOM from both bores, regardless of rainfall period, were dominated by high molecular weight molecules (humic-like fluorophores) associated with the presence of terrestrially derived organic matter (i.e. FI ~ 1.4, $HIX_{EM}>0.9$; [76,78]). Furthermore, most CDOM components (C1-C4) were identified as large molecular weight humic-like compounds derived from terrestrial plant material, with the exception of C5 which was identified as UVA humic-like, a lower molecular weight component that is associated with autochthonous production and microbial processing [74]. The intensity of components C1-C4 was greater in bore D13 for both rainfall periods, which is consistent with the presence of the more modern and less $^{13}C$ enriched terrestrial carbon at this site shown by the isotopic data (i.e. >DOC). Fellman et al. [80] also showed an overall decrease in fluorescence characteristics from the upper to lower catchment in pools of the semiarid Pilbara (Western Australia). The fluorescence results indicate that the dominant source of groundwater carbon at Sturt Meadows aquifer is the terrestrial soil. However, during HR, bore W4 shows elevated BIX (>1.5) values, indicating the presence of CDOM with an autochthonous origin (i.e. microbially derived; [81]), along with an increased relative contributions of a lower molecular weight component (i.e. C5) at this bore. This is again consistent with the

isotopic results, and suggests that the HR event is stimulating specific microbial activity at this site, leading to changes in the recycling of older organic matter, and stable isotopic enrichment.

One potential explanation for this is the infiltration of ions from hypersaline surficial soils into the groundwater during HR, as well as potential mixing with the adjacent lake Raeside (i.e. increased Cl⁻ concentrations during HR, S4 Table), forming a groundwater estuary [39]. Autochthonous CDOM is more common in estuarine and marine environments compared to freshwater bodies [82] and has been reported across microtidal subterranean estuaries [83]. The occurrence of autochthonous CDOM at W4 but not D13 may relate to either its geology (W4 is in calcrete, while D13 has a higher proportion of clay), or its position in the lower half of the bore field which is hydrologically nearer to the neighbouring saline systems. Incorporation of further data from other boreholes across the two geological sectors is needed to fully elucidate the mechanisms underpinning these observations.

One alternative explanation for the CDOM results would be an influx of photochemically altered older carbon from the overlying soils, as the non-mineralized fraction of photobleached CDOM has optical properties that are similar to estuarine and marine CDOM [75,81]. However, there is no obvious explanation as to why this should occur only around bore W4.

Overall, our results suggest that rainfall events play a role in regulating carbon stocks at the SM calcrete, but that the resultant changes are not straightforward. The rainfall events measured were not substantial enough to trigger a full hydrological recharge of the system—something that will become more common with the declining rainfall in the Yilgarn region—but nonetheless sufficiently affected the lower part of the bore field to drive changes in the OM type. To understand the details of this change, a better understanding of the microbiome of the system and its interaction with changes in water chemistry is required. Several investigations have stressed the importance of rainfall events as ecological drivers leading to shifts in biotic community assemblages in groundwater environments [36,84]. The current climate change scenario predicts reduced rainfall events linked to increased droughts, events that are likely to affect the biochemical balance sustaining biota in groundwater [85,86]. Modelling of current ecological dynamics will allow prediction of future effects to the vital (and too often taken for granted) ecosystem services provided by groundwater environments.

## Microbial patterns and carbon pathways

The studied rainfall events triggered shifts in the microbial community assemblages in bore W4. Under both rainfall conditions, the microbial community was typical of saline and hypersaline environments [87,88]. Rhodobacteracea, the most widely distributed bacteria in marine environments [89], was found to be the most dominant family on site. Interestingly, families that were highly abundant under HR (i.e. Phyllobacteriaceae and Rhodospirillaceae) were scarce under the LR period, indicating that rainfall provides conditions for their proliferation; this is again consistent with the findings from the carbon isotopic and fluorescence analyses that there is a change in microbial activity during HR. Conversely, the vast majority of other families–especially Pseudomonadacea, Planococcaceae and Caulobacteraceae–were only present during LR. Genus level analysis indicated a more abundant and diverse community under LR than HR. Decline in biodiversity during recharge events has been ascribed to dilution processes caused by water inflows linked to storm events decreasing the density of micro-organisms and thus their detectability [90], and low recharge regimes have been suggested hosting the baseline autochthonous microbial community [91]. While dilution processes may play a role at SM calcrete, a more comprehensive understanding of the microbial ecological dynamics and their variation over time is needed, requiring further long-term investigation.

The putative assessment of pathways related to cells' carbon metabolisms provided evidence for inorganic carbon fixation and methane pathways (i.e. methane oxidation), two of the most common biochemical routes reported in groundwater systems [32,92,93]. No significant changes in the proportions of each of the main pathways were detected between LR and HR. In a recent study, Hofmann and Griebler [20] tested the 'priming effect'—the activation of microbial growth and OM transformation under increased OM availability—in groundwater. After a series of laboratory experiments under increasing nutrient concentrations, no evidence of priming could be observed. While in overall agreement with these findings, our investigation of specific metabolic pathways revealed a substantial increase in degradative pathways under HR in W4, which is again consistent with the fluorescence and isotopic results. Pathways involved in breaking down aromatic compounds were the most abundant, accounting for 50% (5 out of 10) of the degradative metabolisms that significantly increased after rainfall. Volatile organic compounds, such as toluene, catechol and phenyl acetate, have been found to be very abundant in contaminated aquifers [94,95,96] and may also occur naturally in the hypersaline lakes of Western Australia [97]. Aromatic compounds have been found leaching into groundwater after rainfall and shifting the character of DOM in sandy aquifers [21], confirming their importance as biochemical drivers in typically low energy systems [98]. After rainfall, the microbial community in W4 seemed to profit from increases and changes in OM, as indicated by the high abundances of taxa potentially involved in aromatic compound degradation such as *Stappia* sp., *Roseibacterium* sp., *Tageae* sp. and *Thalassobaculum* sp. [89,99]. Taxa with a high affinity to denitrification processes such as *Paracoccus* sp., *Roseovarius* sp., *Brevundimonas* sp. and *Planomicrobium* sp. [89,100,101], dominated under LR. However, denitrifying *Nitratireductur* sp. was also present under HR, suggesting that nitrogen (nitrates, nitrites and ammonia) provides basal energy sources under both rainfall conditions. However, additional investigations on specific nitrogen pathways of SM calcrete bacteria will be necessary to elucidate this further.

Degradation of glucose (polysaccharide of glucose) also increased under HR, suggesting adaptations to the increased OM. During these conditions, abundances of *Pseudomonas* sp., one of the most opportunistic and versatile bacteria on earth—plummeted, probably due to the repressing effect of glucose on the expression of several genes [102]. The other three degradative metabolic pathways which were more abundant after rainfall, arginine, ornithine and purine, constitute catabolic pathways whose main product is ammonia [103]. High ammonia concentrations were detected under HR [41], previously ascribed to dissolution and overland transport of animal waste [47], and might represent a compendium of nutrient inputs and metabolic production. The only degradative pathway that was significantly overrepresented during LR was the degradation of glycogen, the primary carbon and energy storage compound of many bacteria [104]. Our results are in line with Yamamotoya et al. [105], suggesting that this polysaccharide of glucose is key to long-term bacterial survival and is utilised when carbon sources become limiting, as per the case of the LR period.

Another pathway that followed this decreasing trend after rainfall was the dicarboxylate-hydroxybutyrate cycle. Characteristic for microaerophiles and anaerobes [106], this cycle is considered 'energetically efficient' in contrast to other autotrophic carbon fixation pathways [107]. A plausible explanation is that the dicarboxylate-hydroxybutyrate pathway is activated when OM is scarce (such as LR), and uncommon under HR when OM is more available. In addition to pathways involving OM processing, those involving inorganic carbon fixations, namely the reductive acetyl-CoA pathway and reductive pentose phosphate cycle, also increased after rainfall. Inorganic incorporation might play a role in natural carbon fixation and cycling in groundwater microbes [108], an assumption that has rarely been tested.

## Conclusions

A combination of biochemical and genetic data allowed preliminary untangling of the biochemical function regulating microbial communities at the SM calcrete (Fig 5). Given their importance in allowing the transition between abiotic to biotic frameworks, bacteria are vital in shaping the biochemical flows regulating subterranean biodiversity [109]. However, despite their importance, many questions about subterranean microbial processes remain unresolved. Indeed, the fields of groundwater ecology and subterranean microbiology would mutually benefit from the integration of respective insights. Due to increased natural and anthropic pressures, subterranean biotic communities are currently being exposed to increased losses of taxonomical and functional diversity, leading to poorer and more fragile groundwater ecosystems [110]. Further medium to long term interdisciplinary studies monitoring the changes in groundwater ecological dynamics will allow to assess the impact of climate changes on one of the most essential ecosystems in the world.

## Supporting information

**S1 Table. DOC (Dissolved Organic Carbon) and DIC (Dissolved Inorganic Carbon) concentrations (mg/L), $\delta^{13}$C DOC, $\delta^{13}$C DIC, pMC DOC, $\Delta^{14}$C DOC, Conventional Age DOC, pMC DIC, $\Delta^{14}$C DIC, Conventional Age DIC for the bores W4 and D13.** BP: before present with present being 1950 AD; pMC: percent of modern carbon.
(DOCX)

**S2 Table. Fluorescence/absorbance indices and their definitions.** Adapted from Coble et al. [111]. Note: em = emission wavelengths, ex = excitation wavelengths.
(DOCX)

**S3 Table. Abundances of PICRUSt2 outputs relating to carbon metabolism (KO, level 3) and degradative pathways (MetaCyc).** Pathways in bold indicate the significantly ($P < 0.05$) overrepresented pathways in one of the two rainfall periods.
(DOCX)

**S4 Table. Hydrochemical values of the bores D13 and W4 under LR and HR.** Na: Not available. Units of $\delta^{18}$O and $^2$H in per mil (‰), and units of tritium in TU (Tritium Units).
(DOCX)

**S1 Fig.** Two-dimensional (left panel) and three-dimensional (mid panel) fluorescence landscapes, and the excitation (grey line) and emission (black line) spectra (right panel) for the five different components identified by the PARAFAC model. Intensity is scaled to a maximum fluorescence of 1.
(DOCX)

**S2 Fig. Bar plots illustrating the abundances of genus and families under LR and HR.** Abundances corresponding to the 37 ZOTUs without a reference/belonging to uncultured bacterium were removed from the figure for clarity purposes.
(DOCX)

## Acknowledgments

We wish to acknowledge the traditional custodians of the land, the Wongai people, and their elders, past, present and emerging. We acknowledge and respect their continuing culture and the contribution they make to the life of Yilgarn region. The authors thank Flora, Peter and

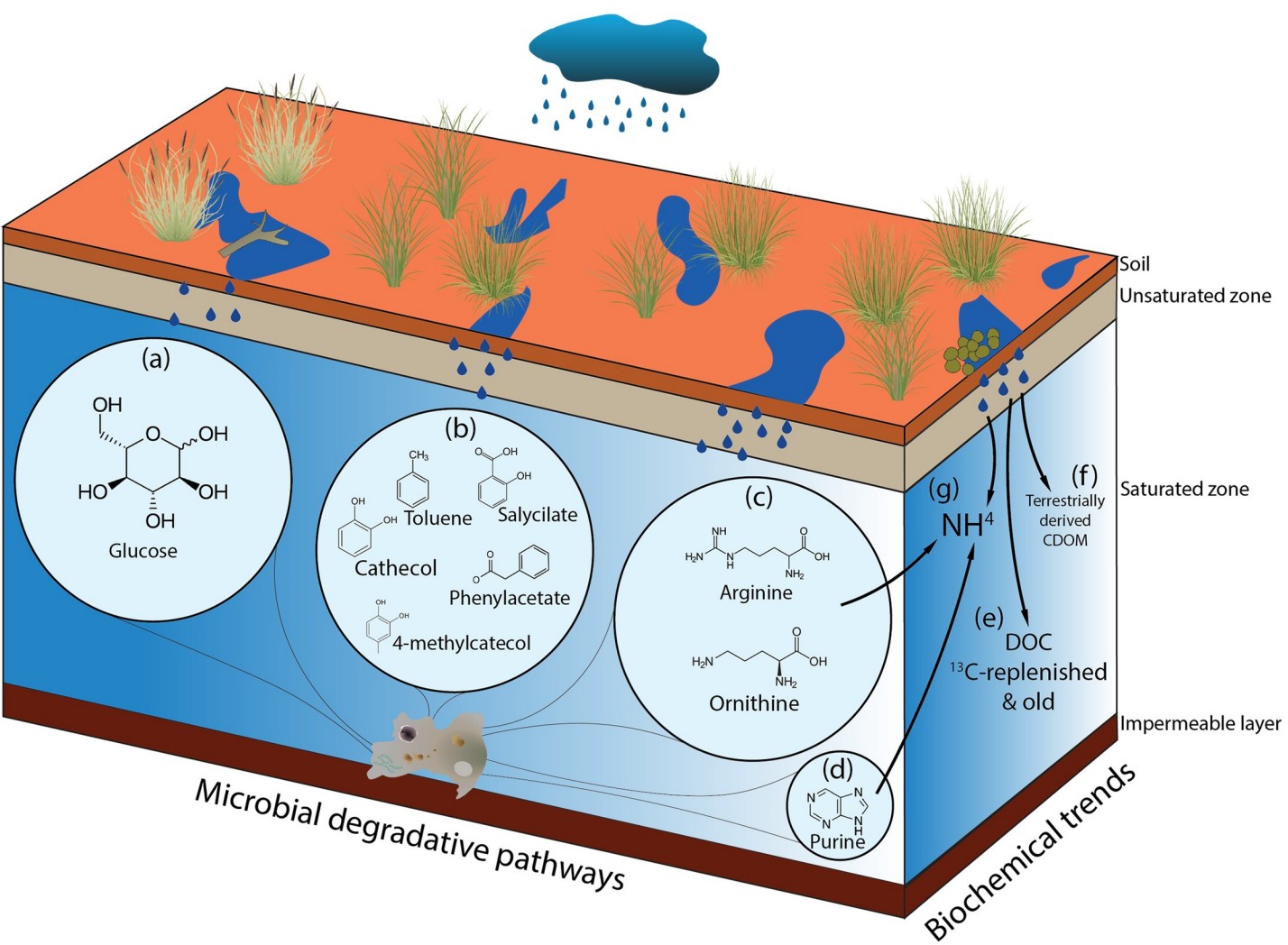

**Fig 5. Scheme of the main degradative pathways and biochemical patterns under HR.** (a) glucose degradation, (b) aromatic degradation, (c) arginine and ornithine degradation, (d) purine degradation, (e) DOC replenishment inferred from isotopic data, (f) terrestrially derived CDOM inflows (fluorescence analysis) and (g) increase in ammonia concentrations as a result of nutrients inputs from the surface and microbial metabolic activities (purine and amino acid degradation).

Paul Axford of the Meadows Station for their kindness and generosity in providing both accommodation and access to their property.

## Author Contributions

**Conceptualization:** Mattia Saccò, William F. Humphreys.

**Data curation:** Mattia Saccò, Jen A. Middleton, Nicole E. White, Matthew Campbell, Masha Mousavi-Derazmahalleh, Alex Laini, Quan Hua, Karina Meredith.

**Formal analysis:** Mattia Saccò, Jen A. Middleton, Nicole E. White, Matthew Campbell, Masha Mousavi-Derazmahalleh, Alex Laini, Quan Hua.

**Funding acquisition:** William F. Humphreys, Nicole E. White, Steven J. B. Cooper, Kliti Grice.

**Investigation:** Mattia Saccò, Alison J. Blyth, Masha Mousavi-Derazmahalleh, Christian Griebler.

**Methodology:** Mattia Saccò, Alison J. Blyth, Jen A. Middleton, Nicole E. White, Matthew Campbell, Masha Mousavi-Derazmahalleh, Karina Meredith.

**Resources:** Alison J. Blyth, Quan Hua, Karina Meredith, Sebastien Allard, Pauline Grierson.

**Supervision:** William F. Humphreys, Quan Hua.

**Validation:** Mattia Saccò, Alison J. Blyth, Steven J. B. Cooper, Sebastien Allard.

**Writing – original draft:** Mattia Saccò, Jen A. Middleton.

**Writing – review & editing:** Alison J. Blyth, William F. Humphreys, Nicole E. White, Matthew Campbell, Alex Laini, Quan Hua, Karina Meredith, Steven J. B. Cooper, Christian Griebler, Sebastien Allard, Pauline Grierson, Kliti Grice.

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
