## [Decision Letter · Decision Letter 0]

21 May 2020

PONE-D-20-08624

Tracking down carbon inputs underground from an arid zone Australian calcrete

PLOS ONE

Dear Dr. Saccò,

Thank you for submitting your manuscript to PLOS ONE. After careful consideration, we feel that it has merit but does not fully meet PLOS ONE’s publication criteria as it currently stands. Therefore, we invite you to submit a revised version of the manuscript that addresses the points raised during the review process.

A rebuttal letter that responds to each point raised by the academic editor and reviewer(s). You should upload this letter as a separate file labeled 'Response to Reviewers'.A marked-up copy of your manuscript that highlights changes made to the original version. You should upload this as a separate file labeled 'Revised Manuscript with Track Changes'.An unmarked version of your revised paper without tracked changes. You should upload this as a separate file labeled 'Manuscript'

We look forward to receiving your revised manuscript.

Kind regards,

Jian Liu

Academic Editor

PLOS ONE

Journal Requirements:

2. We note that Figure 1 in your submission contain [map/satellite] images which may be copyrighted. All PLOS content is published under the Creative Commons Attribution License (CC BY 4.0), which means that the manuscript, images, and Supporting Information files will be freely available online, and any third party is permitted to access, download, copy, distribute, and use these materials in any way, even commercially, with proper attribution. For these reasons, we cannot publish previously copyrighted maps or satellite images created using proprietary data, such as Google software (Google Maps, Street View, and Earth). For more information, see our copyright guidelines: http://journals.plos.org/plosone/s/licenses-and-copyright.

Reviewers' comments:

Reviewer's Responses to Questions

**Comments to the Author**

1. Is the manuscript technically sound, and do the data support the conclusions?

Reviewer #1: Partly

Reviewer #2: Yes

2. Has the statistical analysis been performed appropriately and rigorously? 

Reviewer #1: Yes

Reviewer #2: Yes

3. Have the authors made all data underlying the findings in their manuscript fully available?

Reviewer #1: Yes

Reviewer #2: Yes

4. Is the manuscript presented in an intelligible fashion and written in standard English?

Reviewer #1: Yes

Reviewer #2: Yes

5. Review Comments to the Author

Reviewer #1: In this paper the authors focus on carbon cycling in the exchange of surface water and ground water, using a multi-technique assessment. It is interesting to me, and subterranean carbon flows is an area of ongoing research. The incorporation of functional genomics on bacterial make this study timely, and this topic should be of wide interest. Overall, the paper is well written and structured, and figures are nice. But I have a few concerns about source material and methodology that I would like the authors to address. Below are my specific comments.

1) There are only two bores selected to represent two main geological units (W4 for calcrete and D13 for clayey) and also catchments (D13 for the upper and W4 for the lower). It’s not enough but acceptable. But, why all the samples are collected on two days (on 7/11/2017 as low rainfall treatment, and on 17/03/2018 as high rainfall treatment)? Is it appropriate to use on day to represent the whole high/low rainfall treatment? How many replicates are there (I only found that three sample replicates collected for genetic analyses)? Are these repetitions false repetitions?

2) What is the diameter of bores? Are they usually cover or open (the rainwater can directly fall into)?

3) Did you get any information about 13C and 14C in the rainfall? Precipitation cannot be pure due to the particles in the air, and it may also affect the carbon input.

4) Groundwater samples were collected using a pump, and the bores are 11 m depth (right? From figure 1). How high is the water table at the time of sampling and what layer of the water are you taking? It maybe have big impact on microbes’ composition.

5) Why water sample for functional genomic investigation just collected from bore W4, but not from bore D13 or from both? Is it possible that the differences between the two samples are due to the seasonal dynamics of the microbial community itself? Or to some extent due to?

6) Just a matter of personal opinion, I found the study of microbes in this paper a little jarred, or at least the introduction didn’t make me feel it is compelled to do it.

Line 107: the availability of (OM) which….

Line 170: what are unpredictable recharge dynamics? For example?

Lines 366-368. It’s not a result, and should be place in discussion or methods.

Line 493-496: In the current experimental design, it is not possible to separate the two effects. This is why we need more bores for sampling.

Figure 3a: I don’t think the relative % contribution is needed, its information is also showed in first picture of Fmax.

Table S1: DIC (Dissolved Inorganic Carbon)

Reviewer #2: The authors conducted a comprehensive chemical and microbial analysis of two boreholes in a shallow groundwater system in an Australian desert. Water samples were taken in the dry and the wet season with differing amounts of antecedent rainfall. The purpose was to infer sources and amounts of carbon inputs that form the basis of the aquifer food chain. One borehole was representative of a calcrete section of the aquifer, the other of a clayey section and there were interesting differences between them, indicating different pathways for carbon inputs and differential sensitivity to rainfall. One fascinating result was the intricate shift in the microbial community between the wet and the dry season in the one borehole that received more processed, older carbon as input.

The manuscript is well written, purpose and results are clearly explained and they should be interesting to a broad readership. My only caveat is that some of the specialized terminology of bore holes and bore fields might have been better explained to a broad audience, for example:

151: Explain what capped and lined means and why it is relevant for this study.

163: what is meant by “stabilization of in-field parameters”

93: I am confused about what the “upper” and “lower” part of the bore field is. Does it refer to elevation? If so, is there a relationship between why the one is in calcrete and the other in clay? Might be useful to have a schematic cross section of the unsaturated zone to appreciate the structural differences associated with the functional differences.

The end of the introduction, where you explain the general purpose of the study, you do not explain the relevance of the two boreholes. Since they end up functioning quite differently, it would have been useful to have some kind of introduction here.

6. PLOS authors have the option to publish the peer review history of their article (what does this mean?). If published, this will include your full peer review and any attached files.

Reviewer #1: No

Reviewer #2: No

---

## [Author Response · Author response to Decision Letter 0]

30 Jun 2020

Tracking down carbon inputs underground from an arid zone Australian calcrete 

Ms. Ref. No.: PONE-D-20-08624

Title: Tracking down carbon inputs underground from an arid zone Australian calcrete

Journal: PLOS ONE

Please accept our sincere gratitude for providing constructive comments and suggestions, which, we believe, have greatly improved the paper.

Below, we provide a detailed, point-by-point account of how we responded to each comment

In yellow in the MS: changes according to suggestions from reviewer 1

In light blue in the MS: changes according to suggestions from reviewer 2

In green in the MS: authors’ contributions to an improved version of the manuscript 

Reviewer #1: In this paper the authors focus on carbon cycling in the exchange of surface water and ground water, using a multi-technique assessment. It is interesting to me, and subterranean carbon flows is an area of ongoing research. The incorporation of functional genomics on bacterial make this study timely, and this topic should be of wide interest. Overall, the paper is well written and structured, and figures are nice. But I have a few concerns about source material and methodology that I would like the authors to address. Below are my specific comments.

1) There are only two bores selected to represent two main geological units (W4 for calcrete and D13 for clayey) and also catchments (D13 for the upper and W4 for the lower). It’s not enough but acceptable. But, why all the samples are collected on two days (on 7/11/2017 as low rainfall treatment, and on 17/03/2018 as high rainfall treatment)? Is it appropriate to use on day to represent the whole high/low rainfall treatment? How many replicates are there (I only found that three sample replicates collected for genetic analyses)? Are these repetitions false repetitions?

Thank you for your comments. Rainfall events in the arid Yilgarn region are patchy and highly unpredictable and the study area is located 11 hours by car from our offices and laboratories at Curtin University. While the low rainfall regime does not provide major challenges in terms of collection of samples, sampling campaigns immediately after high rainfall periods are often not possible due to floods and inaccessibility of the area. The sampling design we applied followed the site-specific criteria based on data collected over the last 14 years at Sturt Meadows which is presented in Hyde et al., 2018 (reference [51]). We decided to homogenise the effort between LR and HR sampling events to maximise representativeness of samples and comparison across seasons. We consider that for comparative and investigative purposes, our one-day protocol per rainfall event has robust scientific foundations that conciliate the challenges provided by the study area. Three independent replicates were used for genetic analysis, while statistical comparisons between the parameters of DOC, DIC, δ13CDOC and δ13CDIC were carried out through the analysis of two independent replicates. We agree with the reviewer that more information on this must be included, and we incorporated details in the Line 302. These replicates are independent and the values were obtained from samples collected separately. As a result, we are confident that our statistical design is accurate and reliable. 

2) What is the diameter of bores? Are they usually cover or open (the rainwater can directly fall into)?

Thank you for you notes. We provided further information on the bores as suggested by the reviewer (L152-154). The diameter of the bores is 10 cm and they are capped with PVC to prevent rainwater to fall into the aquifer through the wells.

3) Did you get any information about 13C and 14C in the rainfall? Precipitation cannot be pure due to the particles in the air, and it may also affect the carbon input.

Thank you for your interesting comment. Unfortunately, information about 13C and 14C in the rainfall is not available for area of the Sturt Meadows aquifer. We do agree with the reviewer that these data could provide insights into the hydrogeological patterns observed. The only information available is from the areas of Perth and Rottnest Island, reported by Meredith et al.,2012. Science of the Total Environment, 414, 456-469 and Bryan et al., 2017. Science of the Total Environment, 607, 771-785. Unfortunately, while extremely valuable, we cannot incorporate these data into our study due to the site-specific dynamics shaping these systems. Indeed, future in-detail investigation at Sturt Meadows will highly benefit for the incorporation of the mentioned analysis on rainfall.

4) Groundwater samples were collected using a pump, and the bores are 11 m depth (right? From figure 1). How high is the water table at the time of sampling and what layer of the water are you taking? It maybe have big impact on microbes’ composition.

Thank you for your notes. As specified in the text, bores are unlined, except within about 0.5 m from the surface. As a result, the bores are uncased for most of their length, including for the entire water column. More details on this can be found through the reference [50] provided. The depth of the water table for the bores W4 and D13 is shown in Table S4. Water table at both bores dropped 0.3 m under the HR period: W4 changed from 6.3 m (LR) to 6 m (HR), and D13 shifted from 7.5m (LR) to 7 m (HR). This change is in line with the drop detected in the bore E7 (the bore with the weather station, see reference [41] for further details) and confirms the challenges on defining recharge regimes sensu stricto for these types of subterranean environments. Water samples through pumping were collected from the middle of the water column (at ~3m for bore W3 and at ~3.5m for the bore D13). We agree with the reviewer that microbial composition might change across different layers in the water column. However, our multidisciplinary design involved the feed-back between accuracy and sampling effort (and analytical cost), and we consider that an acceptable compromise that enabled the combination of three novel and usually disconnected analytical designs (isotopes, fluorescence and functional genomics) was reached. Importantly, our approach is in line other interdisciplinary works in groundwaters (e.g. Sang et al., 2018. Scientific reports, 8(1), 1-11; Flynn et al., 2013. BMC microbiology, 13(1), 146.). Nonetheless, aware of the limitations of our study, we tempered our discussion accordingly, suggesting that further studies will be required to address smaller scale dynamics (L536-538; L563-564). However, we consider that our study based on the mentioned ‘triple approach’ (isotopes, fluorescence and metagenomics) provides a valuable and multidisciplinary pilot work that is advancing the field of groundwater ecology. 

5) Why water sample for functional genomic investigation just collected from bore W4, but not from bore D13 or from both? Is it possible that the differences between the two samples are due to the seasonal dynamics of the microbial community itself? Or to some extent due to?

Thank you for your comments. Unfortunately, functional genomics on water samples from the bore D13 was not possible. Whilst a more encompassing strategy is always preferred, as with any project operating within a limited time and budget we focused on just one bore. Environmental conditions in the aquifer are very stable, and rainfall provides a vital ecological driver in the ecosystem. While seasonality might have played a role, further studies to confirm its role are required, as stressed in the discussion (L536-538; L563-564). 

6) Just a matter of personal opinion, I found the study of microbes in this paper a little jarred, or at least the introduction didn’t make me feel it is compelled to do it.

Thank you for your comment. We dedicated an entire section in the introduction on the ecological importance of microbes as baseline drivers of nutrient flows in groundwaters (L108-118). We consider that our results on the trends of microbial functional genomics under contrasting rainfall periods provide interesting insights into the biogeochemical flows sustaining the Sturt Meadows calcrete. In concordance with reviewer 2, we believe that the observed metabolic shifts in the microbial community from the borehole that received more processed, older carbon as inputs are valuable and interesting results. Overall, we consider that the incorporation of community functional aspects into the hydrogeological characterisation considerably helped untangle the ecological functioning of these cryptic systems.

Line 107: the availability of (OM) which….

Thank you for your comment. We amended the error (L108) and improved the following sentence for clarity and consistency purposes (L111).

Line 170: what are unpredictable recharge dynamics? For example?

Thank you for your notes. We improved this section for clarity purposes and also provided an additional reference to help the reader understand the unique hydrological dynamics characterising Western Australian calcretes (L174-175).

Lines 366-368. It’s not a result, and should be place in discussion or methods.

Thank you for your suggestion. We removed this part for the results and left it for the discussion (L479-482).

Line 493-496: In the current experimental design, it is not possible to separate the two effects. This is why we need more bores for sampling.

Thank you for your notes. We included a final sentence of this section to temper our discussion (L501-503).

Figure 3a: I don’t think the relative % contribution is needed, its information is also showed in first picture of Fmax.

Thank you for your comment. We believe that Fig 3b (% contribution) provides a clear visual comparison between rainfall events per each borehole studied, a procedure that would be difficult if we just illustrated Fig 3a. As a result, we prefer maintaining both graphs for clarity and consistency purposes. 

Table S1: DIC (Dissolved Inorganic Carbon)

Thank you for your suggestion. We amended the error.

Reviewer #2: The authors conducted a comprehensive chemical and microbial analysis of two boreholes in a shallow groundwater system in an Australian desert. Water samples were taken in the dry and the wet season with differing amounts of antecedent rainfall. The purpose was to infer sources and amounts of carbon inputs that form the basis of the aquifer food chain. One borehole was representative of a calcrete section of the aquifer, the other of a clayey section and there were interesting differences between them, indicating different pathways for carbon inputs and differential sensitivity to rainfall. One fascinating result was the intricate shift in the microbial community between the wet and the dry season in the one borehole that received more processed, older carbon as input.

The manuscript is well written, purpose and results are clearly explained and they should be interesting to a broad readership. My only caveat is that some of the specialized terminology of bore holes and bore fields might have been better explained to a broad audience, for example:

151: Explain what capped and lined means and why it is relevant for this study.

Thank you for your notes. We included the suggested information in the text for clarity purposes (L152-154). Boreholes are capped with PVC lids to avoid rainfall falling directly into the aquifer. Boreholes were lined with PVC pipes for stability purposes.

163: what is meant by “stabilization of in-field parameters”

Thank you for your comment. The expression “stabilization of in-field parameters” is widely employed in groundwater hydrological studies. It refers to the return of normal baseline conditions after the pre-purging. We included in the text which parameters we monitored (temperature, pH, salinity, dissolved oxygen (DO) and oxidation-reduction potential (ORP)) for clarity purposes and ease the read for a broader audience (L166-167). 

93: I am confused about what the “upper” and “lower” part of the bore field is. Does it refer to elevation? If so, is there a relationship between why the one is in calcrete and the other in clay? Might be useful to have a schematic cross section of the unsaturated zone to appreciate the structural differences associated with the functional differences.

Thank you for your notes. The elevation on site is the same across the boreholes grid. “Upper” refers to northeast and “lower” refers to southwest. To avoid confusion, we included this information both in the abstract (L42-43) and the discussion (L473). We agree that it could be useful to have a schematic cross section of the unsaturated zone. However, the information on the vertical profiles we have available to date is not detailed enough to enable a reliable schematisation of the unsaturated zone, and to our view a partially refined diagram could confuse the reader. We consider that for the comparative purposes of this study, Fig 1 provides a compelling representation of the geological features on site.

The end of the introduction, where you explain the general purpose of the study, you do not explain the relevance of the two boreholes. Since they end up functioning quite differently, it would have been useful to have some kind of introduction here.

Thank you for your notes. Explanation of the sampling design is provided in the methodological section of the study. We consider that the different biogeochemical functioning of the two boreholes is an insight provided by the results of the current study. Our a priori hypothesis involved the null assumption that both boreholes behaved similarly in terms of ecological functioning. As a result, we believe that the incorporation of further details on the relevance of the sampling points considered could confuse the readers and mix the previous knowledge of the system with the findings provided by this study.

Figure 1: we improved the colours of the Fig 1a for clarity purposes.

Figure 5 (renamed Figure 5 NEW): We removed Glycogen from the section ‘(a)’ because the glycogen degradation pathway increases under LR and not viceversa. The goal of this graph is to illustrate pathways that increase under HR.

L5, 23-24 and 615-616: We included the co-author Sebastien Allard due to his contribution with the biogeochemical analyses

L494-495: we amended the grammatical error

L548: we improved the sentence for clarity purposes

L572: we included ‘bacterial’ 

L160-161: We included the copyright sentence as required.

---

## [Decision Letter · Decision Letter 1]

3 Aug 2020

Tracking down carbon inputs underground from an arid zone Australian calcrete

PONE-D-20-08624R1

Dear Dr. Saccò,

We’re pleased to inform you that your manuscript has been judged scientifically suitable for publication and will be formally accepted for publication once it meets all outstanding technical requirements.

Kind regards,

Jian Liu

Academic Editor

PLOS ONE

Additional Editor Comments (optional):

Reviewers' comments:

Reviewer's Responses to Questions

**Comments to the Author**

1. If the authors have adequately addressed your comments raised in a previous round of review and you feel that this manuscript is now acceptable for publication, you may indicate that here to bypass the “Comments to the Author” section, enter your conflict of interest statement in the “Confidential to Editor” section, and submit your "Accept" recommendation.

Reviewer #1: All comments have been addressed

Reviewer #2: All comments have been addressed

2. Is the manuscript technically sound, and do the data support the conclusions?

Reviewer #1: Yes

Reviewer #2: (No Response)

3. Has the statistical analysis been performed appropriately and rigorously? 

Reviewer #1: Yes

Reviewer #2: (No Response)

4. Have the authors made all data underlying the findings in their manuscript fully available?

Reviewer #1: Yes

Reviewer #2: (No Response)

5. Is the manuscript presented in an intelligible fashion and written in standard English?

Reviewer #1: Yes

Reviewer #2: (No Response)

6. Review Comments to the Author

Reviewer #1: (No Response)

Reviewer #2: (No Response)

7. PLOS authors have the option to publish the peer review history of their article (what does this mean?). If published, this will include your full peer review and any attached files.

Reviewer #1: No

Reviewer #2: No

---

## [Editor Report · Acceptance letter]

6 Aug 2020

PONE-D-20-08624R1 

Tracking down carbon inputs underground from an arid zone Australian calcrete 

Dear Dr. Saccò:

I'm pleased to inform you that your manuscript has been deemed suitable for publication in PLOS ONE. Congratulations! Your manuscript is now with our production department. 

Kind regards, 

on behalf of

Dr. Jian Liu 

Academic Editor

PLOS ONE